# Different Metabolites of the Gastric Mucosa between Patients with Current *Helicobacter pylori* Infection, Past Infection, and No Infection History

**DOI:** 10.3390/biomedicines10030556

**Published:** 2022-02-26

**Authors:** Su-Young Son, Choong-Hwan Lee, Sun-Young Lee

**Affiliations:** 1Department of Bioscience and Biotechnology, Konkuk University, Seoul 05029, Korea; syson119@naver.com (S.-Y.S.); chlee123@konkuk.ac.kr (C.-H.L.); 2Research Institute for Bioactive-Metabolome Network, Konkuk University, Seoul 05029, Korea; 3Department of Internal Medicine, Konkuk University School of Medicine, Seoul 05030, Korea

**Keywords:** *Helicobacter pylori*, gas chromatography, gastric mucosa, metabolites

## Abstract

*Helicobacter pylori* (*H. pylori*) alters metabolism during the gastric carcinogenesis process. This study aimed to determine the metabolites in the gastric mucosa according to the status of the *H. pylori* infection. Patients who visited the outpatient clinic for a gastroscopy and *H. pylori* tests were included. Gas chromatography–time-of-flight mass spectrometry (GC-TOF-MS) analysis was performed using gastric biopsied specimens from the corpus. Twenty-eight discriminative metabolites were found in the gastric mucosa of 10 patients with current *H. pylori* infection, in 15 with past infection, and in five with no infection history. The relative abundances (RAs) of amino acids and sugars/sugar alcohols were higher in patients with no infection history than in patients with current or past infection. The current infection group showed higher RAs of organic acids and lower RAs of fatty acids and lipids compared with the other groups. The RA of inosine was highest in the past infection group. Based on GC-TOF-MS analysis findings, metabolites differed not only between the infected and non-infected patients, but also between those with and without infection history. Amino acid and sugars/sugar alcohol metabolites decreased in patients with current or past infection, whereas fatty acid and lipid metabolites decreased only during current infection.

## 1. Introduction

*Helicobacter pylori* (*H. pylori*) infection induces the carcinogenesis process in the human stomach. Some precancerous lesions including gastric atrophy and intestinal metaplasia (IM) may persist even after the regression of *H. pylori*; therefore, the risk of gastric cancer is higher in stomachs with infection history than in *H. pylori*-naive stomachs [1,2]. Furthermore, the risk increases with the degree of atrophy and IM, which extends from the antrum to the corpus [3,4].

*H. pylori* alters metabolism after infection and changes metabolic dynamics when eradicated [5]. *H. pylori* infection leads to the loss of carboxylic acids and amino acids in the corpus and antrum [6]. In the latter study, the corpus exhibited significant metabolite changes related to stress, tissue damage, and nutrient depletion in contrast to the antrum. Furthermore, 43 plasma metabolites involved in amino acid, lipid, and fatty acid metabolism differed between *H. pylori*-infected, non-infected, and gastric cancer patients [7]. Although studies on metabolomic profiling of gastric cancer are increasing, main findings are inconsistent [8]. Moreover, no study has evaluated the differences in metabolism between patients with current infection, past infection, and no infection history.

In this study, we attempted to determine the metabolites found in the gastric mucosa according to the *H. pylori* infection status. We hypothesized that metabolites observed in the corpus would differ not only between the infected and non-infected patients, but also between patients with and without an infection history. Furthermore, we aimed to determine the different metabolites according to the degree of atrophy and IM.

## 2. Materials and Methods

### 2.1. Enrollment of Patients

Patients who visited one outpatient clinic (Dr. Lee S.-Y.) for an upper gastrointestinal endoscopic examination and *H. pylori* tests between March and April 2021 were included. Patients were included if they agreed to undergo a gastric biopsy during endoscopy. Patients were excluded if they had significant comorbidities (renal failure, liver failure, immune deficiency, etc.) or if certain drugs (acid suppressant, antibiotics, antithrombotic agents, immune modulators, etc.) had been administered within 30 days.

All included patients provided written informed consent before the gastroscopy and *H. pylori* tests. This study was approved by the Institutional Review Board (IRB) of the Konkuk University Medical Center (IRB no. 1010691). The procedures performed in this study were in accordance with the ethical standards of the IRB and Helsinki Declaration.

### 2.2. Gastroscopic Examination and H. pylori Tests

The gastroscopy was performed by a gastroenterologist (Dr. Lee S.-Y.) using GIF-H290 (Olympus, Tokyo, Japan). A rapid urease test (PyroPlus, ARJ Medical, Inc., Oldsmar, FL, USA) was performed using gastric biopsy specimens taken from the greater curvature side of the upper body. Pathology findings were reported according to the updated Sydney classification as previously described [9]. Briefly, no infiltration was scored as 0, mild degree as 1, moderate as 2, and marked as 3. *H. pylori* serology and serum pepsinogen (PG) assays were performed as described in our previous study [10].

Endoscopic findings were scored from 0 to 8 based on the Kyoto classification scoring system for gastritis [11]. Gastric atrophy was scored 0 for no atrophy or closed-type I (atrophy confined to the antrum), 1 for closed-type II or III (atrophy exhibiting atrophic border in the lesser curvature side of the body), or 2 for open-type atrophy [12]. IM was scored 0 for no IM, 1 for limited IM confined to the antrum, or 2 for extensive IM observed in both the corpus and antrum. Hypertrophic gastric folds were scored 0 (absent) or 1 (present). Nodular gastritis was scored 0 (absent) or 1 (present). Diffuse redness in the corpus was scored 0 (none), 1 (mild), or 2 (severe).

### 2.3. Confirmation of H. pylori Infection Status

Current infection was diagnosed if the rapid urease test or Giemsa staining findings were positive. Past infection was defined as a successful history of eradication in patients with negative *H. pylori* test findings. The remaining patients without a history of eradication were classified as patients with no infection history. If there was a discrepancy between the *H. pylori* serology and invasive test findings (rapid urease test, histology, and Giemsa staining), the ^13^C-urea breath test (POCone^®^, Otsuka Electronics Co., Ltd., Osaka, Japan) was performed for confirmation.

### 2.4. Sample Preparation for Metabolite Analysis

The extraction of gastric mucosa metabolites was performed using the methods described by Park et al., with a few modifications [13]. Gastric mucosa samples were homogenized and extracted in 100% methanol (400 μL) containing an internal standard solution (10 μL; 2-chloro-phenylalanine, 1 mg/mL in water) using an MM400 mixer mill (Retsch^®^, Haan, Germany) at a frequency of 30 Hz for 10 min with sonication for another 10 min. Subsequently, the extracted samples were incubated for 1 h at 4 °C and centrifuged at 13,000 rpm for 10 min at 4 °C. Then, the supernatants were filtered using 0.2 μm polytetrafluorethylene (PTFE) filters (Chromdisc, Daegu, Korea). The filtered samples were completely evaporated in a speed vacuum concentrator (Biotron, Seoul, Korea). The completely dried extracts were reconstituted with 50 μL of 100% methanol.

### 2.5. Gas Chromatography–Time-of-Flight Mass Spectrometry Analysis

For gas chromatography–time-of-flight mass spectrometry (GC-TOF-MS) analysis, the reconstituted samples (50 μL) were dried using a speed vacuum concentrator. The dried samples were derivatized using the following protocol. First, 25 μL of methoxyamine hydrochloride (20 mg/mL in pyridine) was added to the dried samples and placed on a thermomixer for 90 min at 30 °C to protect the ketone and aldehyde groups. Then, 25 μL of *N*-methyl-*N*-trimethylsilyl trifluoroacetamide (MSTFA), being used as a silylating agent, was incubated for 30 min at 37 °C. One microliter of each derivatized samples was injected into an Agilent 7890A GC system (Santa Clara, CA, USA) equipped with an L-PAL3 autosampler and Pegasus^®^ HT TOF-MS system (LECO Corp., St. Joseph, MI, USA). Metabolites were separated using an RTX-5MS column (30 m length × 0.25 mm inner diameter × 0.25 μm particle size; Restek Corp., St. Joseph, MI, USA) with a constant flow of helium (1.5 mL/min) as the carrier gas. The analytical parameters and program were adopted from our previous study [13]. All samples were run in a randomized manner to reduce systematic errors and bias.

### 2.6. Data Processing and Statistical Analysis

MS data processing and multivariate statistical analyses were performed as described in our previous study [13]. The raw GC-TOF-MS data were converted to netCDF (*.cdf). Then, the .cdf format files were processed using MetAlign software (RIKILT-Institute of Food Safety, Wageningen, The Netherlands) to determine baseline correction, peak selection, peak area normalization, peak mass (*m*/*z*), and retention time (min). Multivariate statistical analysis was performed using SIMCA-P+ software (version 12.0, Umetrics, Umea, Sweden). Principal component analysis (PCA) and orthogonal partial least squares discriminant analysis (OPLS-DA) were performed to compare significantly different metabolites among the three groups. The significance of the OPLS-DA models was determined through the analysis of variance testing of cross-validated predictive residuals (CV-ANOVA) derived from the SIMCA-P+ program. Discriminative variables were selected based on the variable importance in the projection (VIP) scores derived from the OPLS-DA model. The discriminative metabolites obtained from GC-TOF-MS were tentatively identified by comparing their retention time, mass spectrum, and mass fragment pattern with available databases such as the National Institute of Standards (NIST) and Technology database (version 2.0, 2001, FairCom, Gaithersburg, MD, USA), Wiley 9, the Human Metabolome Database (HMDB, http://www.hmdb.ca/ accessed on 19 January 2022), and in-house libraries and standard compounds analyzed under identical GC-TOF-MS analysis conditions. Significant differences were evaluated through ANOVA or the Student’s *t*-test using PASW Statistics 18 software (SPSS Inc., Chicago, IL, USA).

For continuous variables, differences between the three groups were analyzed using ANOVA with Bonferroni correction, and findings were presented as mean ± standard deviation. For continuous variables with asymmetrical distribution, the Kruskal–Wallis test was used, and findings were presented as medians and ranges. Categorical variables were analyzed using the Chi-squared test with Bonferroni correction, and findings were presented as percentage. Furthermore, correlation analysis was performed to verify the link between the RA of each metabolite and the degree of gastric atrophy and IM. The findings were presented as Pearson’s correlation coefficient (*r*) values.

## 3. Results

### 3.1. Infection Status of the Included Patients

Among the 30 included patients, 10 showed positive *H. pylori* test findings and were classified into the current infection group. Fifteen patients with negative *H. pylori* test findings had an eradication history; hence, they were classified into the past infection group. The remaining five patients with negative *H. pylori* test findings were classified into the no infection history group.

### 3.2. Different Test Findings According to the Status of H. pylori Infection

The current infection group showed higher serology titers, higher serum PG II levels, and lower PG I/II ratios than those in the past infection and no infection history groups. The total Kyoto classification score was highest in the current infection group, whereas it was lowest in the no infection history group. The findings of the serum assays and gastroscopy are summarized in Table 1.

### 3.3. Gastric Mucosa Metabolites Observed in Patients with Current H. pylori Infection, Past Infection, and No Infection History

To determine significantly distinguished metabolites among the three groups, non-targeted metabolite profiling of the gastric mucosa was conducted using GC-TOF-MS followed by multivariate statistical analysis, including unsupervised PCA and supervised OPLS-DA. The OPLS-DA score plot obtained from the GC-TOF-MS dataset showed a clearly distinguished pattern among the CI, PI, and NI groups according to OPLS1 (4.96%) (Figure 1A). The statistical model value of OPLS-DA was determined by R^2^X (cum) = 0.316, R^2^Y (cum) = 0.560, Q^2^ (cum) = 0.240, and *p* < 0.05, which indicated the model validation, prediction accuracy, fitness, and cross-validation analysis, respectively. Nonetheless, the unsupervised PCA score plots were not clearly separated from the three groups (Appendix A). Distinguished metabolites among the three groups were identified and selected based on the VIP value (>1.0) from the OPLS-DA model. One-way ANOVA was applied to determine statistical significance (*p* < 0.05).

A total of 28 metabolites including four organic acids, nine amino acids, four sugars and sugar alcohols, six fatty acids and lipids, four others, and one unknown compound were tentatively identified (Table 2). The relative contents of 28 significantly different metabolites were converted into fold changes and displayed on a heat map (Figure 1B). Based on this, the relative abundance (RA) of organic acids, except for pyruvic acid, GABA, salicylic acid, and uric acid, was relatively higher in the current infection group than the RAs in the past infection and no infection history groups. Intriguingly, the RAs of fatty acids and lipids in the current infection group were relatively lower than those in the other two groups. The RA of inosine was significantly higher in the past infection group. The RAs of amino acids (valine, leucine, isoleucine, asparagine, and cystine) and four sugars and sugar alcohols were higher in the no infection history group. Collectively, the current infection group showed higher RAs of organic acids and lower RAs of fatty acids/lipids compared to those in the past infection and no infection history groups. Moreover, the RAs of amino acids and four sugars/sugar alcohols were higher in the no infection history group than those in the other two groups.

### 3.4. Correlation between the Metabolites and Degree of Atrophy and IM

Among the 28 significantly different metabolites, the RAs of valine (*r* = −0.481, *p* = 0.011), leucine (*r* = −0.414, *p* = 0.032), and isoleucine (*r* = −0.404, *p* = 0.038) were inversely correlated with the degree of gastric atrophy. With regard to the degree of IM, only the RA of valine exhibited inverse correlation (*r* = −0.481, *p* = 0.011). The RAs of salicylic acid (*r* = 0.406, *p* = 0.044) and sucrose (*r* = 0.523, *p* = 0.003) showed positive correlations with the degree of IM.

### 3.5. Gastric Mucosa Metabolites Observed in Patients with a History of Gastric Neoplasm or Peptic Ulcer Disease

The included patients did not have active or healing-stage ulcers. Duodenal ulcer scars were observed in two patients in the past infection group. Gastric ulcer scars were observed in four patients owing to previous endoscopic resection for gastric neoplasm. Three patients in the past infection group underwent gastric adenoma resection, whereas one patient in the current infection group underwent early gastric cancer resection. The latter patient developed reinfection after successful *H. pylori* eradication.

Among the 28 metabolites, the RAs of amino acids and sugars/sugar alcohols (except sucrose) were relatively lower in the patients with a history of gastric neoplasm than those in their counterparts (Figure 2A). Conversely, the RAs of amino acids (except GABA and cystine) and inosine were relatively higher in patients with duodenal ulcer scars than those in their counterparts (Figure 2B).

### 3.6. Gastric Mucosa Metabolites Observed in Patients with H. pylori-Negative Gastritis

Inflammatory cell infiltration was observed in all 10 patients with *H. pylori* infection, whereas only seven (46.7%) patients showed mononuclear cell or neutrophil infiltration among the 15 patients with past infection. Among the five H. pylori-naive patients, two (40%) showed a mild degree of inflammatory cell infiltration. Therefore, nine patients were considered to have *H. pylori*-negative gastritis. The RAs of fatty acids/lipids, amino acids (except GABA and cystine), and sugars/sugar alcohols (except sorbitol) were relatively higher in the *H. pylori*-negative gastritis group than those in the *H. pylori* gastritis group (Appendix A).

## 4. Discussion

In this study, we found that metabolites of the gastric corpus differed not only between *H. pylori*-infected and non-infected patients, but also between *H. pylori*-exposed patients (current and past infection groups) and non-exposed patients (no infection history group). From a metabolic perspective, amino acids and sugars/sugar alcohols were decreased in patients with an exposure history. Fatty acid and lipids were decreased only during current infection and seemed to be restored after the regression of *H. pylori*. To the best of our knowledge, this is the first study to show differences in gastric corpus metabolites between the patients with current infection, past infection, and no infection history.

Fatty acid and lipid metabolites decreased in patients with current infection. During *H. pylori* infection, the gastric mucosal barrier becomes fragile due to accelerated fatty acid metabolism in gastric mucosal cells [14]. *H. pylori* infection altered fatty acid metabolism by increasing the concentrations of arachidonic acid and prostaglandin E2 in their study. These changes diminished with the regression of *H. pylori*; hence, decreased fatty acid and lipid metabolites were no longer observed in the absence of infection. Our findings are supported by a previous study showing that lipid peroxide increased during *H. pylori* infection due to gastric mucosal oxidative inflammation [15]. From the perspective of oxidative stress, microvascular leucocyte activation and chemokine and myeloperoxidase activities increased during *H. pylori* infection, with significant changes in lipid metabolites.

In a study using plasma metabolites, the loss of linoleic acid and palmitic acid was correlated with the progression of IM after adjusting for age, sex, *H. pylori* infection, and histological findings [16]. Another study using plasma metabolites of gastric cancer patients found no difference according to the presence of infection, although significant metabolites were found in men and smokers [17]. These two studies suggest that metabolic deregulation of fatty acids and lipids is involved in gastric carcinogenesis; however, they did not find a correlation with current infection. Different from their studies, two Korean studies reported under-expressed lipid metabolites in gastric cancer tissues [18,19]. Because the proportion of current infection, past infection, and no infection is 75.2%, 22.5%, and 2.3% in Korean gastric cancer patients [20], these Korean studies support our results showing that fatty acid and lipid metabolites decrease during *H. pylori* infection. Recent Chinese studies using gastric cancer tissues also reported a significant decrease in lipid metabolites [21,22]. Therefore, when analyzing metabolic profiles, it is important to obtain the gastric tissue and to discriminate patients with and without an infection history.

Notably, metabolites on sugars and sugar alcohols were decreased in patients with current or past infection. In an in vivo study, 11 metabolites of glycolysis, tricarboxylic acid cycle, and amino acid metabolism were altered by *H. pylori* infection [23]. In human studies, several types of glycans showed decreased levels due to altered glycolysis in patients with gastric cancer [24,25]. Similarly, amino acid metabolites were decreased in patients with current or past infection. In an animal study, the glycolytic pathway, tricarboxylic acid cycle, and choline pathway were upregulated, whereas the urea cycle was downregulated after *H. pylori* infection [26].

Gastric cancers are more common in the severe atrophy group than in the moderate, mild, and none atrophy groups [27]. In their study, advanced atrophy, aging, uric acid levels, and ulcers were risk factors of gastric cancer. In our study, the RA of uric acid was increased in the current infection group; however, there was no correlation with the degree of atrophy or IM. Instead of uric acid, valine, leucine, and isoleucine metabolites decreased with the progression of gastric atrophy or IM. Our findings can be supported by a recent study showing that valine, leucine, and isoleucine synthesis and glycolysis are disturbed during gastric carcinogenesis due to the increased energy requirement [28]. Another study on tongue coating metabolites also reported that degradation of valine, leucine, and isoleucine is observed in patients with gastric precancerous lesions [29]. Together with our findings, the downregulation of valine, leucine, and isoleucine seems to be involved with the progression of atrophy or IM.

Interestingly, patients with *H. pylori-*negative gastritis did not show loss of fatty acids, lipids, amino acids, sugars, and sugar alcohols in our study. This finding can be supported by our previous study, which showed that H. pylori-negative gastritis was not associated with atrophy and IM [9]. Only H. pylori-related gastritis can induce endoscopic gastritis, which requires gastric cancer surveillance. Similarly, patients with duodenal ulcer scars showed increased amino acid metabolite levels, whereas patients with a history of gastric adenoma or cancer resection showed decreased amino acid and sugar/sugar alcohol metabolite levels. A recent study showed that the level of metabolites related to glutathione cycle is lower in epithelial cells infected with gastric cancer-derived *H. pylori* strains than those infected with duodenal ulcer-derived strains [30]. They demonstrated that the oxidative stress induced by gastric cancer-derived *H. pylori* strains was stronger than that induced by duodenal ulcer-derived strains. Altogether, patients with *H. pylori-*negative gastritis or duodenal ulcer history rarely show gastric carcinogenesis, in which valine, leucine, isoleucine synthesis and glycolysis are disturbed.

This study had several limitations. First, in an endemic area of *H. pylori* infection, there is a risk of unintended eradication in patients without a history of infection. Older patients may have been infected in the past. Second, our findings may not be observed outside the East Asia, because 94.4–96.2% of Koreans with *H. pylori* infection show positive East Asian type *CagA* findings [31,32]. Third, biopsy samples were obtained only from the greater curvature side of the upper body, because false-negative findings are not rare when samples are obtained from other sites [33,34]. Despite these limitations, we found significant differences in fatty acid/lipid, amino acid, and sugar/sugar alcohol metabolites between patients with current infection, past infection, and no infection history.

## 5. Conclusions

Non-targeted metabolite profiling of gastric mucosa tissues reveals that the loss of amino acids and sugar/sugar alcohols is observed in patients with current or past infection. Loss of fatty acids and lipids seems to recover after regression of *H. pylori*, because it is only observed during current infection. Collectively, gastric mucosa tissue derived from *H. pylori* infection or exposure history affected metabolite variation. Our multi-omics data would be useful for not only determining the status of *H. pylori* infection but also discriminating significant gastritis related to gastric cancer. New insights into delaying atrophy and IM might be provided by understanding the downregulation of valine, leucine, and isoleucine metabolites. Furthermore, the efficacy of novel targeting agents for gastric cancer prevention should be evaluated by measuring the downregulation of the *H. pylori-*related metabolome. Comprehensive multi-omics approaches are needed to understand the meaning of gastric mucosa-related metabolic changes observed during *H. pylori*-induced gastric carcinogenesis.

## Figures and Tables

**Figure 1 biomedicines-10-00556-f001:**
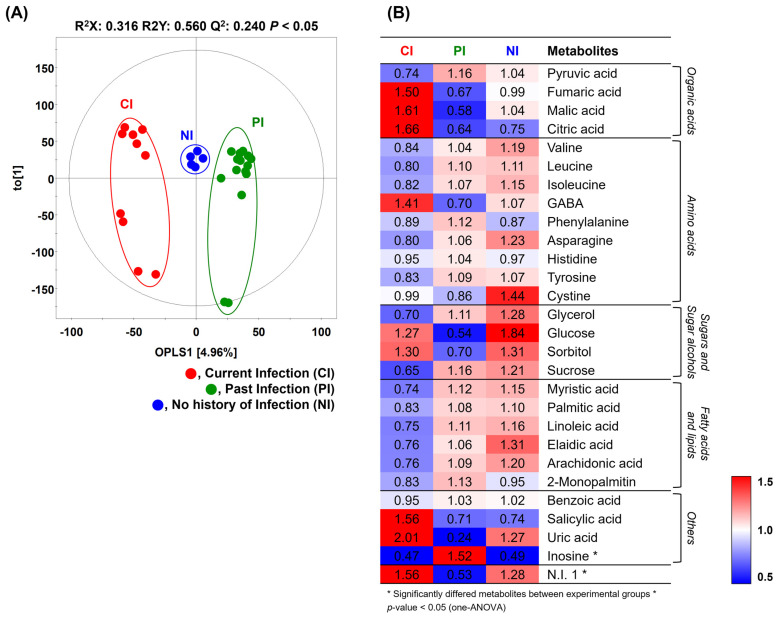
(**A**) Orthogonal partial least discriminant analysis score plots obtained from the GC-TOF-MS dataset of the gastric mucosa samples from three different groups. (**B**) Heat map analysis for the relative abundance of different metabolites (VIP > 1.0) derived from the GC-TOF-MS analysis. The colored squares (blue to red) indicate fold changes normalized by the average of each metabolite. * Significantly different metabolites among the three groups (CI, PI, and NI groups); *p* < 0.05 using one-way ANOVA.

**Figure 2 biomedicines-10-00556-f002:**
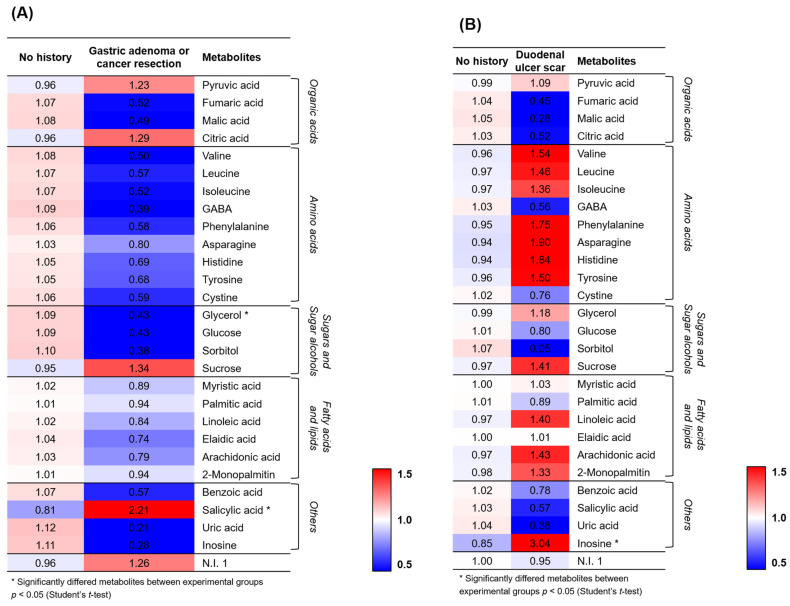
(**A**) Heat map analysis for the relative abundance of different metabolites (VIP > 1.0) derived from the GC-TOF-MS analysis for the comparison between 26 patients without a history of gastric neoplasm and 4 patients with a history of gastric adenoma or cancer resection. (**B**) Heat map analysis for the comparison between 28 patients without duodenal ulcer scars and two patients with duodenal ulcer scars. The colored squares (blue to red) indicate the fold changes normalized by the average of each metabolite. * Significantly different metabolites among the two groups; *p* < 0.05 using the Student’s *t*-test.

**Table 1 biomedicines-10-00556-t001:** Baseline characteristics and test findings of the included patients.

Variables	All Patients (*n* = 30)	Current Infection (*n* = 10)	Past Infection (*n* = 15)	No Infection History (*n* = 5)
Age (years; mean ± SD)	61.6 ± 13.0	66.1 ± 14.2	60.4 ± 10.1 *	56.4 ± 17.7 *
Sex (male)	21 (70%)	7 (70%)	10 (66.7%)	4 (80%)
Serum anti-*H. pylori* IgG (AU/mL)	30.6 (5–200)	117.8 (27.5–200)	6.3 (5–51.3) *	5.4 (5–33.6) *
Serum pepsinogen I (ng/mL)	54.1 ± 18.2	51.8 ± 21.3	57.8 ± 21.8	53.9 ± 12.7
Serum pepsinogen II (ng/mL)	14.4 ± 5.3	18.3 ± 3.9	11.5 ± 3.3 *	8.8 ± 0.5 *
Serum pepsinogen I/II ratio	4.3 ± 2.4	2.8 ± 1.0	5.7 ± 3.4 *	6.1 ± 1.1 *
Updated Sydney system classification scores				
Neutrophil	1.2 (0–3)	2.0 (1–3)	1.1 (1–2) *	0.4 (0–1) *^,^**
Mononuclear cell	0.8 (0–3)	1.0 (0–3)	0.1 (0–1)	0.4 (0–1)
Gastric atrophy	0.9 (0–3)	1.4 (0–3)	1.2 (0–2)	0 *^,^**
Intestinal metaplasia	0.4 (0–3)	1.3 (0–3)	1.1 (0–2)	0 *^,^**
Kyoto classification score for gastritis (sum of 1–5)	2.6 (0–5)	4.0 (3–5)	2.0 (0–4) *	0.4 (0–1) *^,^**
1. Chronic atrophic gastritis score	1.1 (0–2)	1.4 (0–2)	1.2 (0–2)	0.4 (0–1) *^,^**
2. Metaplastic gastritis score	0.5 (0–2)	0.4 (0–1)	0.8 (0–2) *	0 *^,^**
3. Nodular gastritis score	0.1 (0–1)	0.3 (0–1)	0.1 (0–1)	0
4. Hypertrophic rugae score	0	0	0	0
5. Diffuse redness score	0.5 (0–2)	1.9 (1–2)	0 *	0 *
Other significant endoscopic findings				
Gastric ulcer scar	4 (13.3%)	1 (10%)	3 (20%)	0
Duodenal ulcer scar	2 (6.7%)	0	2 (13.3%)	0

* Significant difference compared with the current infection group (*p* < 0.05). ** Significant difference compared with the past infection group (*p* < 0.05). For continuous variables, ANOVA with Bonferroni correction was used, and findings are presented as mean ± standard deviation (SD). For variables with asymmetrical distribution, the Kruskal–Wallis test was used, and findings are presented as medians and ranges. For categorical variables, the Chi-squared test with Bonferroni correction was used.

**Table 2 biomedicines-10-00556-t002:** Distinguished metabolites among the current infection, past infection, and no infection history groups analyzed by GC-TOF-MS.

No.	Ret (min) ^1^	VIP1	Unique Mass (*m*/*z*)	Metabolites ^2^	MS Fragment Pattern (*m*/*z*)	ID ^3^
**Organic acids**						
1	5.87	1.37	235	Pyruvic acid	73, 147, 133, 100, 72, 75, 148, 220, 235	STD/MS
2	7.81	1.50	245	Fumaric acid	73, 147, 99, 245, 241, 75, 79, 52, 113	STD/MS
3	9.10	2.12	233	Malic acid	73, 147, 75, 55, 133, 233, 148, 101, 117	STD/MS
4	11.67	1.45	273	Citric acid	73, 147, 75, 273, 265, 148, 149, 133, 211	STD/MS
**Amino acids**						
5	6.63	1.13	144	Valine	73, 144, 147, 218, 100, 59, 146, 219	STD/MS
6	7.16	1.22	158	Leucine	73, 158, 147, 103, 117, 133, 205, 159	STD/MS
7	7.38	1.27	158	Isoleucine	73, 158, 142, 57, 147, 117, 59, 130	STD/MS
8	9.44	1.35	174	GABA	73, 84, 75, 174, 147, 56, 157, 79, 74	STD/MS
9	10.24	1.33	218	Phenylalanine	73, 218, 192, 100, 147, 219, 193	STD/MS
10	10.56	1.10	116	Asparagine	73, 116, 75, 132, 147, 141, 100, 231, 188	STD/MS
11	12.37	1.18	154	Histidine	73, 157, 79, 147, 52, 74, 58, 117, 218	STD/MS
12	12.46	1.36	218	Tyrosine	73, 218, 100, 219, 75, 179, 147, 220, 280	STD/MS
13	14.66	1.49	218	Cystine	131, 73, 75, 116, 57, 55, 144, 132, 128, 146	STD/MS
**Sugars and sugar alcohols**						
14	7.19	1.78	205	Glycerol	73, 147, 103, 117, 205, 133, 148, 218	STD/MS
15	12.27	1.18	319	Glucose	73, 147, 205, 160, 319, 103, 117, 129, 217	STD/MS
16	12.54	1.67	217	Sorbitol	73, 147, 103, 205, 217, 319, 117, 129	STD/MS
17	16.59	1.53	361	Sucrose	73, 147, 103, 217, 361, 129, 169, 362	STD/MS
**Fatty acids and lipids**						
18	11.75	1.46	117	Myristic acid	73, 75, 117, 132, 129, 145, 131, 285	STD/MS
19	13.05	1.13	313	Palmitic acid	73, 117, 132, 55, 129, 145, 131, 133, 313	STD/MS
20	14.07	1.42	337	Linoleic acid	73, 67, 55, 81, 117, 95, 54, 129, 337	STD/MS
21	14.10	1.16	339	Elaidic acid	75, 73, 117, 129, 67, 96, 84, 145, 339	STD/MS
22	14.99	1.58	91	Arachidonic acid	73, 75, 79, 67, 80, 91, 117, 93, 77, 106, 129	STD/MS
23	15.94	1.29	218	2-Monopalmitin	73, 129, 103, 147, 218, 131, 101, 130, 313	MS
**Others**						
24	6.93	1.27	179	Benzoic acid	79, 52, 73, 147, 58, 135, 116, 171, 179	STD/MS
25	9.31	2.01	267	Salicylic acid	73, 156, 267, 100, 84, 176, 147, 128	STD/MS
26	13.54	1.79	441	Uric acid	73, 147, 217, 191, 305, 318, 133, 129, 441	STD/MS
27	16.17	2.26	230	Inosine ^4^	73, 75, 281, 103, 147, 217, 230, 129, 193	STD/MS
**Unknown**						
28	9.28	1.20	156	N.I. 1 ^4^	73, 156, 75, 147, 157, 84, 158, 100, 230	Not detected

^1^ Retention time. ^2^ Metabolites selected based on VIP (>1.0) values obtained from OPLS-DA model. ^3^ Identification; STD/MS, comparison of mass spectrum with HMDB, NIST, Wiley 9, and in-house library and comparison with standard compounds analyzed under the same condition of GC-TOF-MS. ^4^ Significantly differing metabolites among three different groups (*p* < 0.05 using one-way ANOVA).

## Data Availability

Not applicable.

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
