# Peer review of "Different Metabolites of the Gastric Mucosa between Patients with Current Helicobacter pylori Infection, Past Infection, and No Infection History"

_biomedicines, 2022, doi:10.3390/biomedicines10030556_

Round 1

Reviewer 1 Report

Authors aimed to determine the metabolites in the gastric mucosa according to the status of H. pylori infection. However, this is well designed study and well written manuscript, authors should concern below points.

  1. This study was done in Korea. Because molecular characteristic of H. pylori strain in Asian area is different from it in European area. Therefore, the results may differ if this study was done in European area. Authors should mention this point as one of the limitations.
  2. Authors should discuss more about clinical meaning of this study. For example, could the metabolites be marker of pylori infection status? Please discuss more about this point.
  3. Because H. pylori evokes oxidative stress, elevated lipid peroxide is reported in pylori infection (Aliment Pharmacol Ther. 15(5):715-25, 2001). From the perspective of oxidative stress, authors should discuss more about their results.
  4. Authors should mention whether patient with peptic ulcer or gastric cancer are included in the cohort. If included, authors should evaluate characteristic of these patients

Author Response

1.This study was done in Korea. Because molecular characteristic of H. pylori strain in Asian area is different from it in European area. Therefore, the results may differ if this study was done in European area. Authors should mention this point as one of the limitations.

Thank you for your concern. We added it as a limitation of this study. Please refer to the revised parts highlighted in yellow background.

Lines 311-313, page 10

Second, our findings may not be observed outside the East Asia, because 94.4–96.2% of Koreans with H. pylori infection show positive East Asian type CagA findings [31,32].

2.Authors should discuss more about clinical meaning of this study. For example, could the metabolites be marker of pylori infection status? Please discuss more about this point.

According to your recommendation, we added as follow.

Lines 323-329, page 10

Our multi-omics data would be useful for not only determining the status of H. pylori infection but also discriminating significant gastritis related to gastric cancer. New insights into delaying atrophy and IM might be provided by understanding the downregulation of valine, leucine, and isoleucine metabolites. Furthermore, the efficacy of novel targeting agents for gastric cancer prevention should be evaluated by measuring the downregulation of the H. pylori-related metabolome.

3.Because H. pylori evokes oxidative stress, elevated lipid peroxide is reported in pylori infection (Aliment Pharmacol Ther. 15(5):715-25, 2001). From the perspective of oxidative stress, authors should discuss more about their results.

Thank you for pointing out. We discussed about it as follow.

Lines 256-260, page 9

Our findings are supported by a previous study showing that lipid peroxide increased during H. pylori infection due to gastric mucosal oxidative inflammation [15]. From the perspective of oxidative stress, microvascular leucocyte activation and chemokine and myeloperoxidase activities increased during H. pylori infection, with significant changes in lipid metabolites.

4.Authors should mention whether patient with peptic ulcer or gastric cancer are included in the cohort. If included, authors should evaluate characteristic of these patients.

We added the findings in the bottom of Table 1. We also made a new Figure 2 and mentioned about the findings in the Results and Discussion as follow.

Lines 210-222, page 7

3.5. Gastric Mucosa Metabolites Observed in Patients with a History of Gastric Neoplasm or Peptic Ulcer Disease

The included patients did not have active or healing-stage ulcers. Duodenal ulcer scars were observed in two patients in the past infection group. Gastric ulcer scars were observed in four patients owing to previous endoscopic resection for gastric neoplasm. Three patients in the past infection group underwent gastric adenoma resection, whereas one patient in the current infection group underwent early gastric cancer resection. The latter patient developed reinfection after successful H. pylori eradication.

Among the 28 metabolites, the RAs of amino acids and sugars/sugar alcohols (except sucrose) were relatively lower in the patients with a history of gastric neoplasm than those in their counterparts (Figure 2A). Conversely, the RAs of amino acids (except GABA and cystine) and inosine were relatively higher in patients with duodenal ulcer scars than those in their counterparts (Figure 2B).

Lines 224-230, page 8

Figure 2. (A) Heat map analysis for the relative abundance of different metabolites (VIP >1.0) derived from the GC-TOF-MS analysis for the comparison between 26 patients without a history of gastric neoplasm and 4 patients with a history of gastric adenoma or cancer resection. (B) Heat map analysis for the comparison between 28 patients without duodenal ulcer scars and 2 patients with duodenal ulcer scars. The colored squares (blue to red) indicate the fold changes normalized by the average of each metabolite. * Significantly different metabolites among the two groups; p <0.05 using Student’s t-test.

Lines 297-306, page 9-10

Similarly, patients with duodenal ulcer scars showed increased amino acid metabolite levels, whereas patients with a history of gastric adenoma or cancer resection showed decreased amino acid and sugar/sugar alcohol metabolite levels. A recent study showed that the level of metabolites related to glutathione cycle is lower in epithelial cells infected with gastric cancer-derived H. pylori strains than those infected with duodenal ulcer-derived strains [30]. They demonstrated that the oxidative stress induced by gastric cancer-derived H. pylori strains was stronger than that induced by duodenal ulcer-derived strains. Altogether, patients with H. pylori-negative gastritis or duodenal ulcer history rarely show gastric carcinogenesis, in which valine, leucine, isoleucine synthesis and glycolysis are disturbed.

Again, thank you very much for your helpful suggestions which strengthened our study.

Reviewer 2 Report

The topic of the study is very interesting and worth of exploration. The results contained in the manuscript have a contribution to an interesting field of H. pylori infections and its consequences to the host.  The metabolomic analysis is currently one of the most sophisticated and widely applied method in biological researches. Based on metabolomic patterns one is allowed to perform the comparison of metabolic dynamics in two and more groups of patients leading to first observations which could allow to understand the pathogen-host interactions.

 I have no objections to the quality of the manuscript: methods are appropriate, all parts of the manuscript (introduction, results and discussion) are quite decent. Strength of the study is novel, advanced, deep insight into subject with use appropriate methodology.

Methods: In paragraph of statistical analysis, the authors should add the description provided in Figure1 footnotes concerning the comparison of tested groups.

Discussion: What bothers me a bit is lack of comparison of metabolomic profiles between H. pylori infected patients and patients with gastric inflammation caused by other microorganism.  This could dispel the suspicion that this kind of difference revealed in this study would be found in inflammatory condition regardless the cause.   Moreover, there are studies demonstrating the contribution of gastric microbes in the development and perpetuation of precancerous gastric lesions after H. pylori  eradication. I think that these aspects should be at least discussed by the authors. 

Author Response

1.Methods: In paragraph of statistical analysis, the authors should add the description provided in Figure 1 footnotes concerning the comparison of tested groups.

Thank you for pointing out. We added in the Methods as follow. We also shortened the table footnotes. Please refer to the revised parts highlighted in yellow background.

Lines 133-138, page 3

For continuous variables, differences between the three groups were analyzed using ANOVA with Bonferroni correction, and findings were presented as mean ± standard deviation. For continuous variables with asymmetrical distribution, Kruskal-Wallis test was used, and findings were presented as median and ranges. Categorical variables were analyzed using Chi-square test with Bonferroni correction, and findings were presented as percentage.

2.Discussion: What bothers me a bit is lack of comparison of metabolomic profiles between H. pylori infected patients and patients with gastric inflammation caused by other microorganism.  This could dispel the suspicion that this kind of difference revealed in this study would be found in inflammatory condition regardless the cause. Moreover, there are studies demonstrating the contribution of gastric microbes in the development and perpetuation of precancerous gastric lesions after H. pylori eradication. I think that these aspects should be at least discussed by the authors.

Thanks for your concern. Regarding H. pylori-negative gastritis in patients with past H. pylori infection, we added the results with a new figure S2 as follow.

Lines 231-239, page 8

3.6. Gastric Mucosa Metabolites Observed in Patients with H. pylori-negative Gastritis

Inflammatory cell infiltration was observed in all 10 patients with H. pylori infection, whereas only 7 (46.7%) patients showed mononuclear cell or neutrophil infiltration among the 15 patients with past infection. Among the five H. pylori-naive patients, two (40%) showed a mild degree of inflammatory cell infiltration. Therefore, nine patients were considered to have H. pylori-negative gastritis. The RAs of fatty acids/lipids, amino acids (except GABA and cystine), and sugars/sugar alcohols (except sorbitol) were relatively higher in the H. pylori-negative gastritis group than those in the H. pylori gastritis group (Figure S2).

Lines 339-342, page 12

Figure S2. Heat map analysis for the relative abundance of different metabolites (VIP >1.0) derived from the GC-TOF-MS analysis. The colored squares (blue to red) indicate the fold changes normalized by the average of each metabolite. To determine the statistical significance, p <0.05 obtained from one-way ANOVA was applied. Hp, H. pylori.

Lines 293-306, pages 9-10

Interestingly, patients with H. pylori-negative gastritis did not show loss of fatty acids, lipids, amino acids, sugars, and sugar alcohols in our study. This finding can be supported by our previous study, which showed that H. pylori-negative gastritis was not associated with atrophy and IM [9]. Only H. pylori-related gastritis can induce endoscopic gastritis, which requires gastric cancer surveillance. Similarly, patients with duodenal ulcer scars showed increased amino acid metabolite levels, whereas patients with a history of gastric adenoma or cancer resection showed decreased amino acid and sugar/sugar alcohol metabolite levels. A recent study showed that the level of metabolites related to glutathione cycle is lower in epithelial cells infected with gastric cancer-derived H. pylori strains than those infected with duodenal ulcer-derived strains [30]. They demonstrated that the oxidative stress induced by gastric cancer-derived H. pylori strains was stronger than that induced by duodenal ulcer-derived strains. Altogether, patients with H. pylori-negative gastritis or duodenal ulcer history rarely show gastric carcinogenesis, in which valine, leucine, isoleucine synthesis and glycolysis are disturbed.

Again, thank you very much for your helpful suggestions which strengthened our study.

Round 2

Reviewer 1 Report

Well revised manuscript. No further comments.